# `Synatra`: Turning Indirect Knowledge into Direct Demonstrations for Digital Agents at Scale

**Tianyue Ou**♠    **Frank F. Xu**♠    **Aman Madaan**◇    **Jiarui Liu**♠    **Robert Lo**♠
**Abishek Sridhar**♠    **Sudipta Sengupta**♣    **Dan Roth**♣    **Graham Neubig**♠    **Shuyan Zhou**♠
♠ Carnegie Mellon University    ♣ Amazon AWS AI    ◇ xAI
{tianyueo, fangzhex, gneubig, shuyanzh}@cs.cmu.edu

## Abstract

LLMs can now act as autonomous agents that interact with digital environments and complete specific objectives (*e.g.,* arranging an online meeting). However, accuracy is still far from satisfactory, partly due to a lack of large-scale, *direct* demonstrations for digital tasks. Obtaining supervised data from humans is costly, and automatic data collection through exploration or reinforcement learning relies on complex environmental and content setup, resulting in datasets that lack comprehensive coverage of various scenarios. On the other hand, there is abundant knowledge that may *indirectly* assist task completion, such as online tutorials that were created for human consumption. In this work, we present `Synatra`, an approach that effectively transforms this indirect knowledge into direct supervision at scale. We define different types of indirect knowledge, and carefully study the available sources to obtain it, methods to encode the structure of direct demonstrations, and finally methods to transform indirect knowledge into direct demonstrations. We use $100k$ such synthetically-created demonstrations to finetune a 7B `CodeLlama`, and demonstrate that the resulting agent surpasses all comparably sized models on three web-based task benchmarks Mind2Web, MiniWoB++ and WebArena, as well as surpassing `GPT-3.5` on WebArena and Mind2Web. In addition, while synthetic demonstrations prove to be only 3% the cost of human demonstrations (at $0.031 each), we show that the synthetic demonstrations can be more effective than an identical number of human demonstrations collected from limited domains.[1]

## 1    Introduction

AI agents that operate within a digital environment (*e.g.,* a browser in a computer) intelligently to accomplish complex tasks (*e.g., "Create my July online shopping expense report."*) have the potential to improve the productivity across a broad swath of tasks performed by humans every day [2, 33, 6, 58]. However, agents still lack the ability to complete tasks with a high degree of reliability, partly due to a paucity of training data for such agentic tasks. Typically, supervised finetuning is a standard way to adapt large language models (LLMs) to tasks such as text generation or classification, large-scale demonstration collections for digital agents are not readily available.

For AI agents, demonstrations typically involve specifying a sequence of actions and observations that results in successful task completion, as shown on the right of Figure 1. Existing works that automatically collect demonstrations (1) set up environments for an agent to interact with, (2) run a baseline agent within this environment, and (3) employ scoring functions to remove low quality demonstrations [9] or perform relabeling [1, 26]. All three of these requirements limit applicability to a variety of practical applications. Setting up an environment that is representative of the actual

---

[1]Data and code are publically available at https://ooottttyyy.github.io/synatra

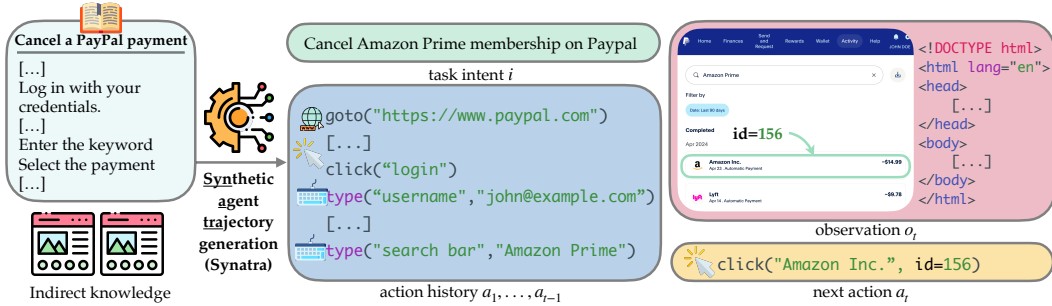

Figure 1: Our approach aims to synthesize direct demonstrations (right of the arrow) that specify the immediate next actions based on previous actions and current observations. The sources comprise only indirect knowledge (left of the arrow), such as tutorials designed for human consumption and randomly sampled observations that lack associated tasks and actions.

environments in which we would like agents to act is a difficult task, and existing environments for digital agents are generally limited in scope to a few websites or digital apps [58, 7, 45]. Even within these constrained settings, strong LLMs such as GPT-4 struggle on tackling tasks [58], making collecting successful demonstrations with LLMs inefficient. In addition, human annotation is costly and its scope can still be limited [17, 6]. For example, gathering a demonstration for canceling a PayPal order requires an actual PayPal account with a legitimate subscription history.

In this work, we propose Synatra, a data generation approach that synthesizes high-quality trajectories for complex digital tasks at scale. This approach is based on the intuition that there is a rich trove of existing knowledge that encodes *indirect* supervision about how to perform digital tasks (§2.2). One example of a piece of indirect knowledge is a tutorial that details the sequential breakdown of a complex web task, such as "how to cancel a recurring payment on Paypal" for human readers (Figure 1 upper left) [53, 57]. While this tutorial provides some procedural guidance, such as "Enter the keyword," it does not specify the executable actions or the tangible observations associated with them. The key idea of Synatra is that, given this indirect knowledge, we can leverage an LLM to *re-purpose* it into more usable form that directly demonstrates the exact action to take given a concrete observation (§3). In doing so, we leverage LLMs' ability to paraphrase and code, as well as its general knowledge of how tasks are performed (Figure 1). The main benefit of this approach is that it can scale with the availability of indirect knowledge created for humans, rather than rely on direct human annotations or LLM trajectories.

We carefully study the sources of indirect knowledge, the design of demonstration formats, and the mechanisms for iterative refinement in order to synthesize high-quality demonstrations using an LLM (§3, §4). We generate demonstrations of $100k$ tasks from 21 domains, and finetune a 7b Codellama-instruct model with this synthetic data. The resulting agent, Synatra-CodeLlama, surpasses existing open-source models of similar size, excluding those finetuned with human annotations, on three web-based task benchmarks: Mind2Web, MiniWoB++, and WebArena. Moreover, it consistently outperforms models that are ten times larger and have been finetuned with interactive data (§6.1). Our findings also indicate that our model is on-par or a more accurate option in browser copilot settings, in comparison to GPT-3.5. Importantly, while each synthetic example incurs only approximately 3% of the cost of a human-annotated demonstration, we demonstrate that synthetic data with good domain coverage can be *more* effective than an identical quantity of limited-domain human demonstrations in real-world web-based tasks in WebArena (§6.2).

## 2 Problem Formulation

### 2.1 Controlling Digital Agents through Natural Language

An agent interacts with a computer environment $\mathcal{E} = \langle \mathcal{S}, \mathcal{A}, \mathcal{O}, \mathcal{T} \rangle$ with state space $\mathcal{S}$, action space $\mathcal{A}$, observation space $\mathcal{O}$ and the environment dynamics $\mathcal{T} : \mathcal{S} \times \mathcal{A} \longrightarrow \mathcal{S}$. While this framework can be applied to different types of task, in this work, we consider a *web browser* as the unified entry point to access different web applications. We follow WebArena [58] to define the observation space as the contents on screen, represented as in text-based accessibility tree format. We use the same universal action space as in WebArena. This action space resembles the keyboard and mouse operations of a

computer (*e.g.,* `click`, `type`), and is applicable to *arbitrary* web applications. Appendix A lists all valid actions. The environment dynamics (*e.g.,* the effect of clicking a button) and the states (*e.g.,* the database status) are decided by the implementation of each web application.

Given the natural language intent $i$, at each time step $t$, an agent issues an action $a_t \in \mathcal{A}$ based on $s_t$. The environment state is updated to $s_{t+1}$ with new observation $o_{t+1}$. This process ends when the agent predicts the `stop` action. We follow existing works [6, 47, 58, 54] to represent $s_t$ with current observation and all previous actions in the form of $(i, a_1, ..., a_{t-1}, o_t)$. A benchmark (*e.g.,* WebArena) supplies a scoring function $r(i, s_n)$ that examines the final state and returns $1$ if the desired goal state is satisfied, and $0$ otherwise.

## 2.2 Definition of Direct Demonstrations and Indirect Knowledge

We consider the expected action $a_t$ given $s_t$ as a form of direct demonstration, *i.e.,* $(s_t, a_t)$. This allows an agent to directly learn how to predict the next action under a given state. On the other hand, indirect knowledge is broadly defined as resources that can benefit the task execution, but is not in the format of state and expected action tuple. We mainly focus on three types of indirect knowledge:

1. **Procedural knowledge** details the sequence of steps $\langle a'_1, a'_2, ..., a'_n \rangle$ required to complete a specific task $i$. Unlike an action $a_t$ in trajectories, the steps in procedural knowledge are ungrounded, they lack a direct association with any particular observation and are not tied to specific action spaces. For instance, the tutorial in Figure 1 instructs the user to *"login with your credentials"* without providing the concrete PayPal login page and the input fields to `type`.

2. **Environment knowledge** $\mathcal{T}$ that describe the effects of applying different actions in some hypothetical states. Example knowledge includes verbal descriptions such as *"... after clicking the cancel button, you will see a pop up window ..."*.

3. **Ungrounded observations** $o$ that are not associated with particular tasks or trajectories. In the context of web-based tasks, an observation could be any random web page with different content and status (*e.g.,* a product page with a query in the search field).

# 3 Scalable Demonstration Synthesis for Digital Agents

In this section, we first introduce our design choices on the canonical formalization of trajectories, which account for the *structural* nature of procedures. Then, we delve into the sources for acquiring indirect knowledge, and the mechanisms for re-purposing this knowledge into direct supervision.

## 3.1 Trajectories as Programs

Existing works demonstrate that representing task-related procedures as programs is beneficial for several reasons. This includes benefits from the structural nature of programs compared to free-form text [56, 23], and the flexibility of using tools [4, 10, 41]. Inspired by these observations, we represent a Python function that interleaves natural language planning articulated in comments and actions as API calls, as shown on the right. The pink background represents the prompt, while the blue background corresponds to the model's response format.

```
website = "<url>"
observation = "<AXtree of page>"
objective = "[...]"

# past actions
def solve():
    # sub-task 1: [...]
    # <action NL explanation>
    action(arg=value)
    [...]
    # sub-task 2: [...]
    [...]
```

```
    # <action CoT reasoning>
    action(arg=value)
    # <action summary>
```

The planning process includes both task-level planning, which breaks the task into multiple sub-tasks, and action-level planning, which explains the low-level goals of each executable action. The model's generation consists of chain-of-thought (CoT, [43, 47]) reasoning that analyzes the objective, previous actions, and current observations. Since CoT reasoning includes detailed information about the current step that may not be relevant for future steps, we further design the model response to include an action summary. This summary serves as a description of the predicted action that is added

to the prompt for future action predictions. A concrete example can be found in Appendix B.[2] We study the empirical effect of program formalization in §6.3.

## 3.2 Synthesizing from Text Procedural Knowledge with Generative Environment

The Internet offers fairly extensive procedural knowledge that describes *how* to perform high-level tasks by breaking down the task into detailed lower-level steps, such as how-tos and tutorials.

**Source** We use wikiHow[3] as our main source for these tutorials due to its comprehensive coverage of diverse tasks as well as its consistent format. Each article consists of a high-level task description and step-by-step instructions. We performed a filtering step and only kept the articles that involve navigation through the graphical user interface (GUI) of a computer or a mobile phone. We prompted `GPT-3.5-turbo` with six examples mixing the target articles (*e.g.,* How to redeem an Amazon gift card online[4]) and non-target articles (*e.g.,* How to make a pizza) to perform the classification of all wikiHow articles. The prompt is shown in Appendix C. As a result, we obtained $25k$ articles that can be used to perform data synthesis.

**Synthesis Approach** We want to bridge two gaps to re-purpose $\langle a_1', a_2', ..., a_n' \rangle$ for task $i$ into $\langle a_1, ..., a_{t-1}, o_t \rangle$. When re-purposing a sequence of actions $\langle a_1', a_2', ..., a_n' \rangle$ for task $i$ into a new sequence $\langle a_1, ..., at-1, o_t \rangle$, many challenges arise. First, the action descriptions provided in tutorials are not constrained to specific action spaces. Instead, they are presented as free-form natural language (NL) expressions, which can lead to ambiguity. For instance, various verbs such as "enter," "input," and others may all correspond to the same underlying action, `type`. Second, NL descriptions are often abstract, omitting concrete actions. For example, the process of "logging in" involves a series of actions, including typing in a username and password, but these specific actions may not be explicitly mentioned. Finally, the steps outlined in tutorials are ungrounded, meaning they are not directly associated with observable states or outcomes. Tutorials typically employ generic descriptions to accommodate various instances of conceptually similar tasks. For example, as illustrated in Figure 1, the tutorial merely instructs to "enter the keyword" without addressing any specific scenario.

Based on these findings, we propose an *iterative* approach that first uses an LLM to rewrite an article into a hypothetical trajectory in the format shown in §3.1, then we leverage a generative model to synthesize the intermediate observation between two consecutive actions. First, in the rewriting step, we ask the assistant LM to perform: (1) propose a hypothetical concrete scenario relevant to the task (2) perform basic parsing such as translating "enter the keyword [...]" into `type(``search bar'', ``Amazon Prime'')`; (3) categorize actions into groups that reflect the sub-task structures outlined by coding blocks. These tasks mainly demand a LLMs's creativity, language processing ability, and event understanding respectively. An example of rewriting a how-to article into a trajectory in program format is showed in Appendix E. The prompt for the rewriting step is in Appendix D.

Because the previous procedures still only result in a sequence of ungrounded actions, we next leverage the assistant LM to generate the observations between randomly sampled consecutive actions. We use the consecutive actions of `type(``search bar'', ``Amazon Prime'')` and `click(``Amazon Inc'', id=156)` in Figure 1 as an example. There are mainly two requirements for generated observations. First, the observation must reflect the *outcomes* of past actions. In the example, this corresponds to a page with a user logged in, and a search input field filled with "Amazon Prime". Second, the observation *encodes* the necessary elements to perform the next action. In the example, this corresponds to a payment history list with a payment to Amazon. We prompt the assistant LM with the action sequence to generate a HTML snippet that fulfills the above requirements. Since the next action requires the concrete element to interact with, we ask the model to insert a tag of `id=``next-action-target-element''` in the corresponding HTML node to indicate the grounding.[5] This step mainly requires a model's coding capabilities, particularly in web development. However, we find that it is not necessary for the LLM to generate HTML with high fidelity and

---

[2]Some existing works also study using program formalization for web-based tasks [11, 46], but focus on action-level interventions without incorporating task-level planning.
[3]https://www.wikihow.com/Main-Page
[4]https://www.wikihow.com/Apply-a-Gift-Card-Code-to-Amazon#Online
[5]The tag is removed through post-processing.

complexity, which is a open research question [36]. The full prompt is in Appendix D and an example for this step is in Figure 5.

Note that in addition to wikiHow, there are other resources that share similar a similar procedural flavor, such as the captions of YouTube how-to videos [25]. Our transformation mechanism is generally applicable to such resources, but we leave concrete examination of them as future work.

### 3.3 Synthesizing from Random Observations

While the procedure in the previous section results in *real procedures*, the LLM-based method generates *simplified observations*. To compensate for this, we also perform data synthesis with *real observations*, and use synthesis to generate *simplified procedures*. We show that these two sources can compensate each other by examining the generated data in §4 and comparing the actual web-based task performance in §6.3.

**Source**  We utilize ClueWeb [27] as our data source, which comprises HTML snapshots of more than 10 billion web pages. Our initial analysis indicates that a random sampling approach would likely lead to a homogeneous distribution dominated by less interactive pages, such as news articles. In contrast, more complex web-based tasks typically require interactions with various web elements to advance the tasks. To diversify the sampled web pages, we employed a temperature sampling approach to select pages based on their content categories. In general, web pages from higher frequency top-level domains in ClueWeb are typically more interactive, such as Amazon and Reddit, while domains with lower frequency are less interactive, such as news article. We use a temperature sampling with $T = 0.6$ so that the sample probability of choosing a page in domain $i$, $P_i' = p_i^{\frac{1}{T}} / \sum_k p_k^{\frac{1}{T}}$, where $p_i$ is the original probability of choosing a page in domain $i$, and $k$ is all the available domains. In doing so, we up-sample more interactive sites while maintaining diversity on more rare sites. More details are listed in Appendix J.

**Synthesis Approach**  We treat each sampled web page as an intermediate observation at time step $t$, aiming to synthesize the specific task $i$, the previous actions $a_1, ..., a_{t-1}$ and the subsequent action $a_t$ consisting of an action, a corresponding target element in the observation, and a natural language summary of the action. We first convert a web page into its corresponding accessibility tree at the beginning of each node, and sample a segment to present to the assistant LM. We follow the WebArena convention of assigning a unique ID to each node in the tree to ease the challenge of referencing the nodes. To increase the diversity of the tasks, we first instruct the model to brainstorm $k$ task categories relevant to the web domain. Then the model randomly selects three of these categories and develops them into concrete scenarios with past actions leading up to the current observation and the next action to take. The prompt is in Appendix F and an example generation is in Appendix G.

### 3.4 Data Filtering

To ensure the quality of the training set, we apply a two-part filtering pipeline. In the first part, we ensure that data samples are both complete and coherent. For a sample to pass this filter, it must include all required components in the correct format. These components include: (1) an action that falls within the defined action space, (2) a valid and meaningful action target element in the corresponding web page, (3) NL texts that are well formed, e.g. without the use of "..." in the texts as an abbreviation by the generative model, (4) overall comprehensive generation without placeholders from the prompt (*e.g., <brief step description>*).

To further eliminate accurately formatted but unresponsive actions, we apply a second filtering step using next state prediction. Here, we use the LLM to predict the next state, $o_{t+1}$, based on the current state $o_t$ and action $a_t$ in our synthesized data. If the model predicts that $o_{t+1} = o_t$, the action is deemed to have no impact, and we filter out the action accordingly.

## 4  Data Statistics

We assess the distribution of history lengths, task objectives, and observations in the form of accessibility trees. The first statistic reflects general task complexity, as longer trajectories typically indicate

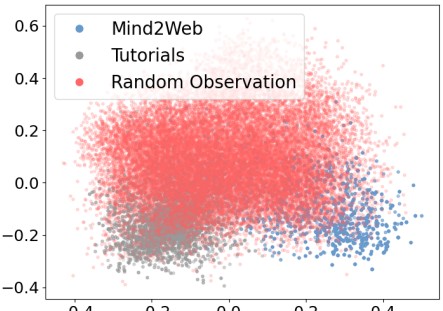
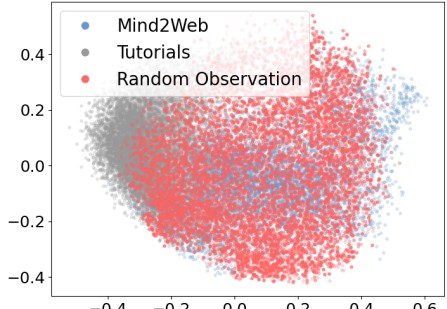

Figure 2: Left: t-SNE of task intent embedding. Right: t-SNE plots of accessibility tree embeddings more complex tasks. The latter two measure the domain diversity of our synthetic data. We also present these statistics for the Mind2Web dataset as an example of human-collected trajectories.

As shown on the right, the majority of our synthetic data consists of trajectories with a history length of fewer than eight steps, regardless of the source. Fewer trajectories have longer histories, with the longest exceeding 20 steps. Examples with longer histories represent more complex scenarios such as *"Create a new YouTube ad campaign for the Summer Collection with a focus on the 25-34 age group interested in sports, fitness, fashion, and style"*. In addition, we analyze the distribution of generated intents by projecting their high-

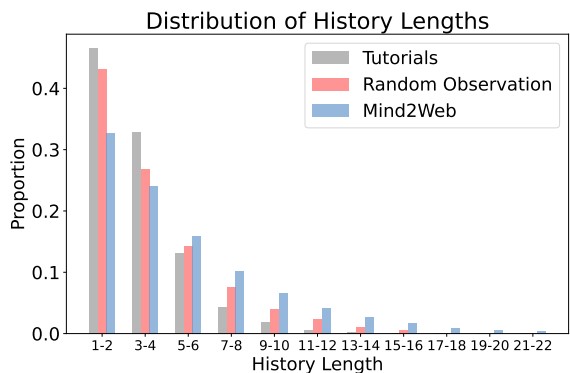

dimensional vectors using the embedding model `all-mpnet-base-v2` [30]. We visualize these embeddings with t-SNE [40]. As shown on the left of Figure 2, intents synthesized from random observations exhibit broader coverage compared to those from tutorials. This may be because humans often prefer to write tutorials for critical domains, while randomly sampled observations offer a more objective reflection of the diverse internet. Although our data synthesis process is entirely independent of Mind2Web, the generated intents show substantial coverage of Mind2Web tasks, which were created by human annotators from top-use websites. Finally, we apply the same embedding and visualization approach for accessibility trees and display the results on the right of Figure 2. The generated observations from tutorials overlap significantly with real web pages, both from random observations and those in Mind2Web.

**Cost**    The average end-to-end cost per sample is $0.031, out of which $0.028 is used to generate the actual sample, and $0.003 is used by our prediction-based filtering pipeline. These values represent the final average cost to generate one valid sample, including the additional cost for samples that were generated but later filtered out, which we distributes across all non-filtered samples.

## 5    Experimental Setup

**Agent training**    We finetune `CodeLlama-7b` [31] with $99,920$ web-navigation instruction examples, sourced from WikiHow articles and ClueWeb web pages [6]. Training details can be found in Appendix K.

**Evaluation tasks**    We select three evaluation tasks in the domain of web navigation. (1) the Mind2Web test set [6]. It consists of three categories, namely, cross-task, cross-website, and cross-domain—ranked by their degree of deviation from the training data. Since our model is not trained

---

[6]We select `CodeLlama-7b` after comparing `Llama-2`, `Llama-3`, `Llama-3-Instruct`, `CodeLlama`, `CodeLlama-Instruct`, `DeepSeek Coder`, and `Mistral` by finetuning and evaluating on Mind2Web's train and test set.

Table 1: Performance of various models in three web-based task benchmarks. We measure step accuracy (%) for Mind2Web, and task success rate (%) for MiniWoB++ and WebArena. The numbers of `FireAct-7b` is taken from [5]; `AutoWebGLM-7b(S1)` represents the model trained with only synthetic data in [19]. All other numbers are produced by our work.

| Model | Mind2Web | MiniWoB++ | WebArena |
|---|---|---|---|
| | Single step Reference-based | Short Execution-based | Long Execution-based |
| *API-based Models* | | | |
| GPT-3.5 | 12.79 | 39.57 | 6.16 |
| GPT-4 | 29.09 | 53.04 | 14.41 |
| *Open-source Instructed Models* | | | |
| CodeLlama-instruct-7b | 6.62 | 23.04 | 0.00 |
| Llama3-chat-8b | 11.50 | 31.74 | 3.32 |
| Llama3-chat-70b | 22.27 | 48.70 | 7.02 |
| *Open-source Interactive Data Finetuned Models* | | | |
| FireAct-7b [3] | - | - | 0.25* |
| AgentLM-7b [51] | 2.99 | 15.65 | 0.86 |
| CodeActAgent-7b [41] | 3.13 | 9.78 | 2.34 |
| AutoWebGLM-7b(S1) [19] | - | - | 2.50* |
| AgentFlan-7b [5] | 3.80 | 20.87 | 0.62 |
| Lemur-chat-70b [46] | 14.28 | 21.30 | 2.95 |
| AgentLM-70b [51] | 10.61 | 36.52 | 3.07 |
| Synatra-CodeLlama-7b | 15.85 | 38.20 | 6.28 |

on the Mind2Web training set, all categories are treated as held-out sets. Therefore, we report an overall step accuracy as the average across all examples. In addition, we simplify the two-stage setup used in Mind2Web, where the model first selects the top 50 candidate elements from the HTML before predicting the action on a selected element. Instead, we input in-viewport accessibility trees that include the ground-truth element. This approach is more challenging because it introduces more irrelevant elements, but it streamlines the prediction process by eliminating the need for an additional element selection model. (2) MiniWoB++ [21]. It is an interactive benchmark with simplified web pages and low-level tasks (*e.g.,* "Enter BUL to the box"). On average, completing a task in MiniWoB++ requires fewer steps than that of WebArena. We tested on a text-only subset of tasks used in [51, 41]. To get the final score, we take the average score of five runs. (3) WebArena. In WebArena, agents are required to navigate in real-world web applications to accomplish high-levels web-based tasks (*e.g.,* "How much I spent last month on grocery"). Both MiniWoB++ and WebArena offer outcome-based evaluations, using programmatic checkers to validate key features after execution and assess whether the task has been successfully completed.

**Baselines** We compare the performance of our model with 12 baseline models, including API-based models such as `GPT-4`, open-source instructed models like `CodeLlama`, and models finetuned with interactive data. For instance, `Agent-LM` [51] is finetuned on supervised data for tasks such as web navigation and interactive coding. Since most baseline models are not optimized for code generation, we follow Zhou et al. [58] and prompt the models to generate natural language responses instead of programs. The same prompt, providing general web navigation instructions without dataset-specific explanations, is used across three datasets. We show the prompt in Appendix H. Notably, we adopt the original prompts of `AgentLM` and `CodeActAgent` [41] when evaluating them on MiniWoB++ to be consistent with their original evaluations. We further remove the task-specific in-context examples to ensure a fair comparison with other models in zero-shot settings.

# 6 Results

## 6.1 Main Results

Table 1 presents the performance of various models across three web-based task benchmarks. Overall, `Synatra-CodeLlama` achieves the best performance among models of comparable size. Notably,

Table 2: Error rates (%) on five fine-grained metrics.

| Error Type | GPT-4-turbo | Llama3-chat-70b | Llama3-chat-8b | CodeLlama-Instruct-7b | Ours |
|---|---|---|---|---|---|
| Nonexistent element | 0.56 | **0.37** | 2.40 | 94.70 | 0.70 |
| Invalid actions | **1.75** | 2.17 | 9.74 | 58.88 | 4.91 |
| Click non-clickable | 5.34 | **3.40** | 15.20 | 100.00 | 5.77 |
| Type non-typable | **1.01** | 1.34 | 11.97 | 32.62 | 4.34 |
| Repeated type | 26.85 | 44.72 | 32.14 | 46.32 | **6.79** |

`Synatra-CodeLlama` significantly outperforms its base model `CodeLlama-instruct-7b`. While `CodeLlama-instruct-7b` fails to complete any tasks in WebArena, `Synatra-CodeLlama` successfully executes 6.28% of them. Furthermore, `Synatra-CodeLlama` elevates the performance of `CodeLlama-instruct-7b` from 6.62% to 15.85% in Mind2Web (a 139.42% relative improvement) and from 23.04% to 38.20% in MiniWoB++. More encouragingly, `Synatra-CodeLlama` demonstrates superior performance on Mind2Web and WebArena compared to `GPT-3.5`. It also outperforms `Lemur-chat-70b`, which is finetuned with interactive data and is ten times larger, across all three benchmarks. The results suggest that our data synthesis approach is effective in helping the model predict the next action (as in Mind2Web) and performing simple tasks with a few steps (as in MiniWoB++). The synthesized data can also guide the model towards executing real-world complex tasks more accurately.

`Synatra-CodeLlama` surpassed the performance of all open-source model finetuned with interactive data. Among these models, `AgentLM`, `CodeActAgent` and `AgentFlan` include demonstrations to perform web-based tasks in their instruction finetuning dataset. However, we find that these models may not serve as capable agents to perform web-based tasks due to the special design choice encoded in the finetuned models. For instance, `AgentLM` and `CodeActAgent` use Regex expression to match interactive element on a web page and require carefully selected in-context examples to showcase which are the proper Regex expression for different examples. However, Regex expressions only work for simple web pages with a few elements as in MiniWoB++, while it is prohibitive to do pattern matching in complex web pages as in Mind2Web and WebArena. As a result, when we experiment with the more generic action space which is suitable for all three benchmarks without in-context examples, we see these models have a significant performance degradation. On the other hand, `Synatra-CodeLlama` targets at the generic web-based tasks and does not encode any dataset artifacts during training. Even though all three benchmarks are completely held-out during data generation, `Synatra-CodeLlama` achieves consistent superior performance on all benchmarks.

## 6.2 Analysis

**What does the model learn from the synthetic data?** Our analysis suggests that the baseline acquires essential knowledge at multiple levels from synthetic data. The error rates for different error types are shown in Table 2. At the low level, synthetic data helps the model *emit valid actions*. For example, while approximately 95% of actions predicted by the base model `CodeLlama-instruct-7b` involve interacting with non-existent elements on the web page, `Synatra` reduces this error to just 0.7%. This low error rate is closer to that of larger models such as `Llama3-chat-70b` and `GPT-4`, which are better at following complex instructions with multiple requirements. Additionally, the synthetic data improves the model's ability to *understand web pages* more accurately. To assess this, we measure the ratio of invalid actions, including both invalid `click` and `type`, where the predicted action attempts to interact with an element that is not clickable (*e.g.,* a piece of text) or not typable (*e.g.,* a button). We observe that models in the 7b–8b range have high error rates in interpreting basic web elements. For instance, the strong 8b model `Llama3-chat-8b` mistakenly `click` a non-clickable element over 15% of the time, while `Synatra` reduces this error to under 6% approaching the performance of `GPT-4`. Finally, at task completion level, `Synatra` shows a stronger ability to track progress accurately. Specifically, when filling out forms on web pages, `Synatra-CodeLlama` is 74% less likely than `GPT-4-turbo` to repeatedly input the same words into the same field, a common error in models. This indicates that our synthetic data enables the model to more precisely recognize completed actions and identify the remaining steps needed to achieve the goal. More qualitative study can be found in Appendix I.

**Can `Synatra` be as effective as human annotations?** We compare models trained with $9k$ human demonstrations (human only) from Mind2Web [6] and $9k$ demonstrations generated by `Synatra`. The

results are shown in Figure 3. Simply training `CodeLlama` with the Mind2Web human annotations provides modest improvement of 3.96% on MiniWoB++ , while failing entirely on WebArena.

In contrast, `Synatra` led to a substantial improvement of 17.81% on MiniWoB++ and 4.56% on WebArena. The limited performance of the human annotations can be partially attributed to their restricted task coverage: notably, Mind2Web lacks information seeking tasks that require a string answer. Furthermore, the human trajectories did not specify conditions for terminating task execution. Despite adding such trajectories from `Synatra` (human + synthesis), the model still under-performed.

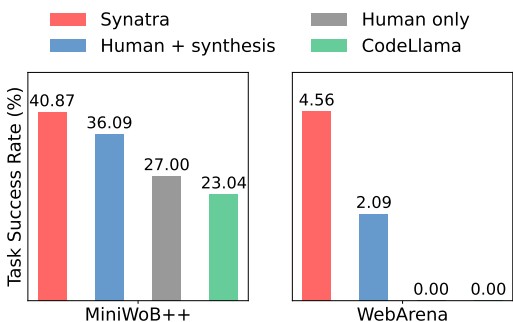

We hypothesize the diversity of tasks plays a role in this discrepancy, since many tasks cannot be covered without pre-set environment (*e.g.,* return an order) (Figure 2). These findings underscore the efficacy of `Synatra`, which also

Figure 3: the comparison between the models trained with trajectories generated by our approach and the data collected from human.

exempts from the complexities of developing recording tools and managing communication with annotators. However, it is important to recognize that the quality of human demonstrations is presumably higher, but they require meticulous design and control during the data collection process.

## 6.3 Ablations

In this section, we perform an ablation to validate the design choices of our data synthesis approach.

**Model performance improves with more synthetic data**
To assess the impact of scaling our synthetic data, we trained three `CodeLlama` 7B models with $18k$, $50k$, and $100k$ samples respectively, while keeping all other parameters constant. We then evaluated the models on WebArena. As shown on the right, the success rates on WebArena steadily increase as the synthetic training data scales from $18k$ to $100k$ samples. This highlights the potential of scaling synthetic data with our approach.

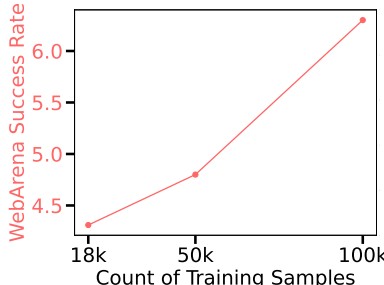

**Representing trajectories as programs is beneficial**   To verify if the programs format is helpful, we convert $30k$ trajectories to the NL format similar to setting in WebArena [58] and compare its performance with model trained with the exact data, but in our program format. The results are shown in Figure 4a. We can see that performance drops on both MiniWoB++ and WebArena when using the NL format. We hypothesize that program, in the form of API calls, is potentially a more natural format to represent sequence of actions. Our observation also echo the observations of using program representation for non-programming tasks [23, 29, 41], while our experiments further contributes insights towards finetuning setups for interactive tasks.

**Different sources of indirect knowledge complement each other**   Our indirect knowledge comes from two sources: tutorials and randomly sampled web pages. In the former source, the procedural knowledge from tutorials are written and verified by human. However, there is no guarantee on the authenticity of the generated observations of web pages. In contrast, the observations from the latter sources are completely real, while there is no guarantee of the trajectory accuracy. We hypothesize that the two sources can compensate each other. To test this hypothesis, we trained three models: one using $9k$ synthetic data mixed from both sources, and two others each using $9k$ data exclusively from one of the sources and the results are shown at Figure 4b. We observe a noticeable performance degradation when models are trained with data from only one source. This indicates that utilizing multiple sources yields a more comprehensive dataset by integrating the precise procedural knowledge from tutorials with the realistic observations of web snapshot data.

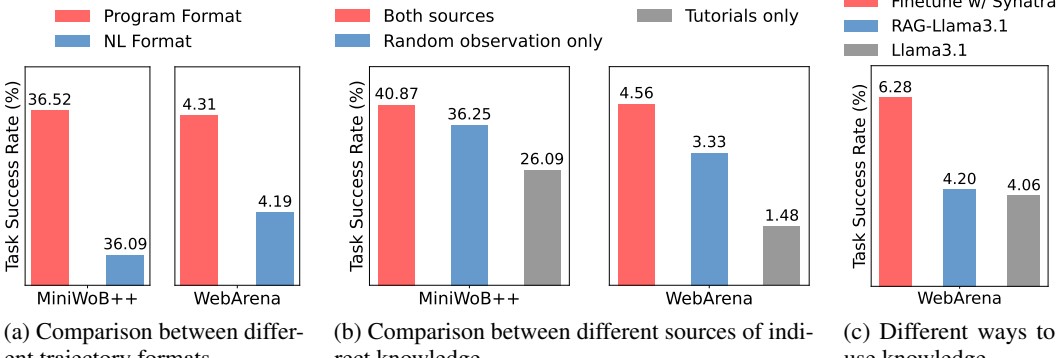

(a) Comparison between different trajectory formats.

(b) Comparison between different sources of indirect knowledge.

(c) Different ways to use knowledge.

Figure 4: ablation on trajectory formats, sources of knowledge, and ways to use knowledge.

**Models have difficulty learning from indirect sources alone**    To access if turning knowledge into trajectories is helpful, we tested a retrieval-augmented approach that directly uses the collection of tutorials as the knowledge base. We first projected text tutorials to embeddings with `all-mpnet-base-v2` [30]. Then a retriever retrieves the most relevant three tutorials measured by cosine similarity. These tutorials were included as additional context in the prompt. We tested this approach with `LLama3.1-instruct-8B`. As shown in Figure 4c, feeding in tutorials directly improves model performance on WebArena marginally , and finetuning on our data shows a substantial advantage. This comparison indicates that models have difficulty making use of indirect knowledge to solve agent tasks, and trajectories is a preferable format for agent learning.

# 7    Related Work

**Learning from Indirect Supervision**    Due to the costly nature of human supervision, many digital agent learning works explore learning from existing yet indirect knowledge sources [8, 35, 12, 55], reinforcement learning optimization that learn from environment feedback [37, 21, 32, 39]. More recently, LLM self-improvement during inference time [34, 49, 28, 44, 14, 52] has been applied on creating more complex digital agents in the wild. Our work fills the gap of training with synthetic data generated from existing resources, *i.e.,* indirect supervision, on complex web navigation tasks.

**Prompting Approaches for AI Agents**    Existing methods include performing reasoning about the current statues before proceeding to next actions [43, 47, 22], search and planning [48, 13], and self-verification [18, 34, 24]. Our focus on instruction tuning data generation without human supervision can hopefully enable synthetic data generation recipes for various kinds of instruction prompts for agent tasks.

**Data Generation for Interactive Agents and Instruction Finetuning**    Existing works design ways of generating training data that adapt LLMs to agent-specific tasks [11, 19], while our work aim to generate more realistic data without human intervention. Considering more broadly over all instruction finetuning: [59, 50, 38, 16] generate instructions of a specific task given a few examples. [42, 15] generates task-agnostic large-scale instruction tuning data without given examples. Li et al. [20] adopts an iterative instruction data generation and model finetuning approach, While our work generates instruction tuning datasets from readily available, indirect, unstructured knowledge resources.

# 8    Conclusion

We propose a data synthesis approach `Synatra`. Empirically we showcase that finetuning with data generated from our approach can improve existing general-purpose LLMs to perform better on digital agent tasks. Since `Synatra` can synthesize trajectories given a single, static piece of indirect knowledge, we argue that when equipped with a capable LLM, a regular tutorial designed for human consumption and a random observation, can also be re-purposed a trajectory. We show that even considering the relative high cost of calling state-of-the-art LLMs such as `GPT-4`, synthesizing is more cost-effective than collecting human demonstrations of similar quantity for model training.

## Acknowledgements

This research project has benefitted from the Microsoft Accelerate Foundation Models Research (AFMR) grant program, and a grant from Amazon Web Services. We would also like to thank Center of AI Safety (CAIS) and CMU FLAME center for compute support.

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

# Table of Contents in Appendix

## A Action Space

Table 3: Action Space of WebArena

| Action Type | Description |
|---|---|
| noop | Do nothing |
| click(elem) | Click at an element |
| hover(elem) | Hover on an element |
| type(elem, text) | Type to an element |
| press(key_comb) | Press a key comb |
| scroll(dir) | Scroll up and down |
| tab_focus(index) | focus on $i$-th tab |
| new_tab | Open a new tab |
| tab_close | Close current tab |
| go_back | Visit the last URL |
| go_forward | Undo go_back |
| goto(URL) | Go to URL |

## B Trajectory Representation

```
website = "<url>"
observation = "<AXtree of the page>"
objective = "Find and review the estimated value of your property on the website."

# past actions
def solve():
    # sub-task 1: Look up your property on Zillow
    # step 1: Search for your address on Zillow's homepage search
    bar to open the property page.
    type(element_id="6135", string="Main St")
    # step 2: From the property details page, navigate to the "Zestimate" section.
    scroll(down)

    # sub-task 2: Begin adjusting the estimated value
    # step 3: Click on 'Edit home facts' to adjust details that
    might affect the home's estimated value.
    click(element_id="9945")
```

## C Prompt to Filter wikiHow Articles

**Prompt to filter wikiHow articles**

Given the title of an article, determine if it is about performing a task solely with computer or mobile phone's graphical user interface, and without any physical world configurations.

input: How to Set Up Chromecast WiFi (Using an Android Phone or Tablet)
output: Set Up Chromecast WiFi involves both a mobile application and physical interactions with the Chromecast device such as plug in the device, so the answer is "No"

input: How to Change Your Desktop Wallpaper on Linux Mint (Using the Linux Mint Wallpapers)
output: Linux Mint is a desktop operating system, and changing the desktop wallpaper is typically done through the system settings or desktop environment's configuration tools, which are desktop applications, so the answer is "Yes"

input: How to delete a file using command line in Linux
output: Command line interface (CLI) in Linux is a text-based interface not a graphical user interface (GUI), so the answer is "No"

input: How to Reboot an iPad (Frozen iPads)
output: Rebooting an iPad usually involves physical actions like pressing and holding buttons on the iPad, so the answer is "No"

input: How to Connect the Kindle Fire to the
Internet (Connecting to an Existing Wi-Fi Network)
output: Kindle is neither a computer nor a mobile phone, so the answer is "No"

input: How to Pair AirPods to an iPhone (If Your AirPods Won't Connect)
output: Pairing AirPods with an iPhone typically includes physical actions such as opening the AirPods case near the iPhone and possibly pressing a button on the AirPods case, so the answer is "No"

input: {{Title of the article}}
output: {{Model prediction}}

# D  Prompt to Generate Demonstrations from Tutorials

## Prompt to rewrite an article to a trajectory in program format

# Task overview
You are given an article about performing a task in a web browser. Your goal is to make this article as accessible as possible to a user who is not familiar with the functionalities of the websites and the task at all.

# Guideline
Read the article carefully and follow the instructions below:
- Assume you start with the home page of the web application, skip the initial 'goto' action.
- Break down the article into a sequence of steps.
- In every step, provide a concrete example that reflects a real execution. This example should clearly describe the element you are interacting with, the concrete value of an element you select, the precise content you type and other details. Never use broad descriptions. The example should be creative and realistic, avoid boilerplate text such as email@example.com. Make sure that the example is consistent across steps.
- Following the concrete example, provide the Python API call corresponding to the example.
- Group all API calls into multiple sub-sections, each section corresponds to a logical and actionable sub-task.

There are special scenarios and here are the ways to deal with them: - If the article describes multiple scenarios or multiple ways to approach the same goal, you can use your own judgement to choose the most common one to describe. - If there are repeated steps, make sure to unroll the steps and describe each of them canonically. - Always assume you perform this task using a web browser, if the original article uses a desktop app or mobile phone app, simply assume the corresponding web app exists. Hence, any steps regarding installation or login can be skipped.

# APIs
The APIs are as follows: 'click(element_desc: str)' - Click on an element.
'element_desc' is the the displayed text or the most representative attribute of the HTML element.
'hover(element_desc: str)' - Hover over an element.
'click_and_type(element_desc: str, content: str)' - Click an input element and type 'content' into it.
'key_press(key_comb: str)' - Press a key combination. 'key_comb' is the combination of keys you want to press on. The default OS is MacOS if there is no explicit specification in the article.
'goto(url: str)' - Navigate to 'url'
'go_back()' - Go back to the previous page.
'go_forward()' - Go forward to the next page.

'new_tab()' - Open a new tab.
'close_tab()' - Close the current tab.
'switch_tab(tab_index: int)' - Switch to the tab with index 'tab_index', starting from 0.

# Response format
Your response should follow the following format.
'''python
 sub-task <index>: <sub-task description>
 # step <index>: <the real execution with concrete values for each argument>
<API, do not skip the keys in the API calls>

 # step <index>: <the real execution with concrete values for each argument>
<API, do not skip the keys in the API calls>

<repeat for all sub tasks>

 # task: <task command given to a smart assistant, only the necessary details on expectation are
needed.>
'''

# Article
{{Article here}}

---

## Prompt to generate observation for two consecutive actions

# HTML Background Knowledge
Commonly used interactable elements in HTML:
['a', 'button', 'input', 'textarea', 'select', 'option', 'label', 'form', 'details', 'summary', 'map', 'area',
'iframe', 'embed', 'object', 'dialog', 'menu', 'fieldset', 'legend', 'datalist', 'output', 'progress', 'meter',
'keygen']

# Task Overview
You are given:
- A browser-based task
- A seuqnece of past actions to perform the task and
- The next action to perform the task.

Your goal is to recover the HTML and the dynamic of a web application with the following requirements:
- The web page embodies a same level of content richness as advanced web applications on the internet. That
is, the web page should have around 80 elements and at least 20 interactable elements. The depth of the
DOM tree should be around 7. The length is at least 3000 tokens.
- Analyze the past actions and determine which of these actions have visible or functional impacts on the web
page you design. Reflect the effects of these past actions in your HTML code. This may involve updating
text, adding new elements, or modifying the layout or styles to represent the state of the web page after these
actions.
- Design the interactable element that enables the next action. Make sure the choice of element type,
attributes, and other essential characteristics are correct. For example, a text field is not interactable. Once
the element is designed, assign the attribute id="next-action-target-element" to this interactable element.
- Please focus on making the static HTML visually rich. Ignore CSS animations & style and JavaScript
functionality in your HTML code. - Provide the concrete reason to perform the next action.

# Response format
'''html
<HTML that fullfils the requirements, make sure 'next-action-target-element' is always included>
'''
<Summarize the progress by analyzing past actions. Provide a brief reason for performing the next action.
 Keep it short. Use imperative sentences.>

```
# Provided information
task: {{task description}}

past actions:
```python
{{past actions}}
```

next action:
```python
{{next action}}
```
```

# E  Example Synthetic Demonstration from Tutorials from a wikiHow Article

## A wikiHow article

How to Use Google Chat on iPhone (Enabling Google Chat in Gmail)

Open Gmail on your iPhone or iPad. This app icon looks like a white and red envelope. If you don't have it, you can download the Gmail app for free from the App Store.

Tap and tap Settings. The three-line menu icon is in the top left corner of your screen, and the option is generally at the bottom of the menu next to a gear icon. If you have multiple Gmail accounts, select the one you want to use Google Chats with. You can repeat these steps if you want to use Google Chats with multiple Gmail accounts.

Tap to check the box next to "Chat. A checked box indicates that Google Chat is active in your Gmail account and you'll see the Chat and Spaces tabs near the bottom of your screen. If Google Chat is enabled, you can tap these tabs to move between conversations in Chat and your emails in Gmail. If this is turned off, you'll need to use the Google Chats app to see your conversations in Chat.

## The output from the rewriting step

```
# sub-task 1: Navigate to Settings to activate Google Chat in another account
# step 1: Click on the menu icon to reveal options
click(element="Menu Icon")

# step 2: Go into Settings by clicking on it
click(element="Settings")

# step 3: Select another Gmail account, this time for jane.doe@gmail.com
click(element="jane.doe@gmail.com")

# sub-task 2: Activate Google Chat in the chosen account
# step 4: Click to enable Google Chat by checking the box
click(element="Check box next to Chat")

# step 5: Ensure the Chat feature is activated
stop(answer="Google Chat activated for jane.doe@gmail.com")

# task: Enable Google Chat on web version of Gmail for jane.doe@gmail.com
```

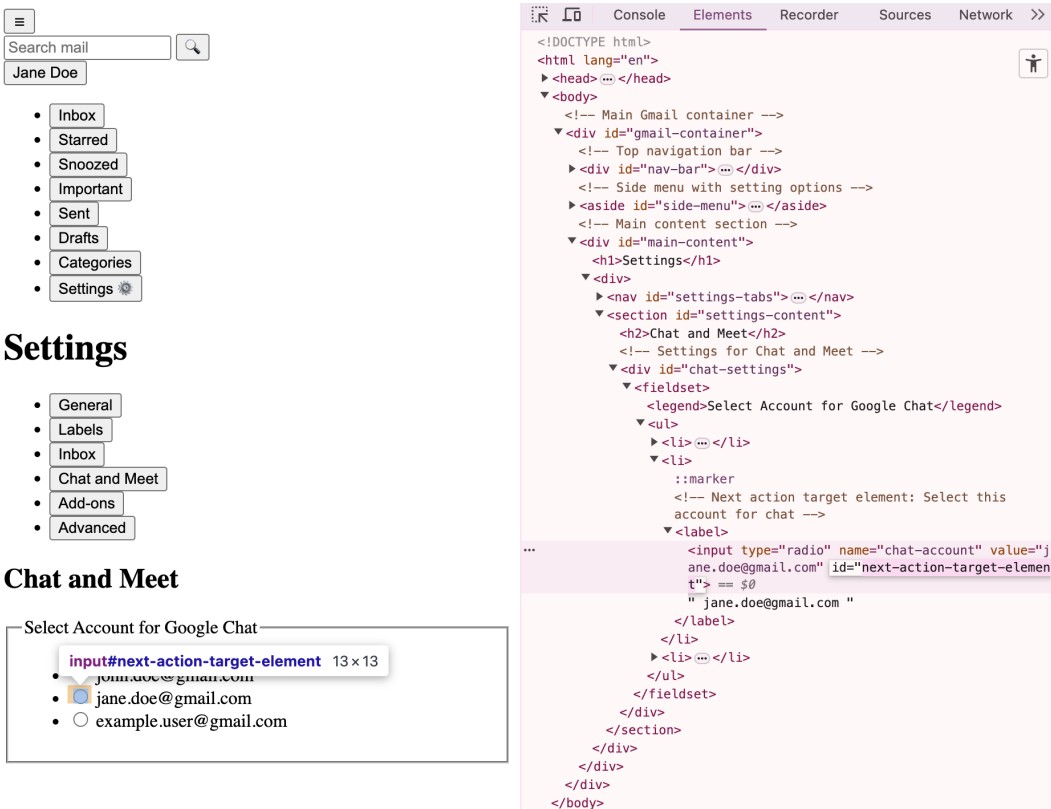

Figure 5: The rendered generated HTML between step 2 and step 3, where step 2 is goto(''Google Chat'') and step 3 is click(''jane.doe@gmail.com''). The concrete element to interact with is tagged with id="next-action-target-element".

## F    Prompt to Generate Direct Demonstrations from Random Observations

---

**Prompt to Generate Direct Demonstrations from Random Observations**

## Task overview
Given the accessibility tree of a web page, your goal is to propose creative and diverse browser-based tasks that involves interacting with this page, along with the previous actions that lead to the current state and the next action needed to be taken to accomplish the task.

## Action space
Here are the allowed actions that you can take to interact with the web page:
'click(element: str, element_id: int=0)' - Click on an element. 'element' is the displayed text or the most representative attribute of the HTML element. 'element_id' is the index of the element at the beginning of the node.
'click_and_type(element: str, content: str, element_id: int=0)' - Click and type 'content' into an 'element'.
'key_press(key_comb: str)' - Press a key combination. 'key_comb' is the combination of keys you want to press on. The default OS is MacOS if there is no explicit specification in the article.
'goto(url: str)' - Navigate to 'url'
'go_back()' - Go back to the previous page.
'go_forward()' - Go forward to the next page.
'new_tab()' - Open a new tab.
'close_tab()' - Close the current tab.
'switch_tab(tab_index: int)' - Switch to the tab with index 'tab_index', starting from 0.
'scroll(up|down)' - Scroll the page up or down.
'stop(answer: str='')' - The task is completed. If the task is to seek information, include the answer as a string. Otherwise, leave it empty.

## Guidelines
You will follow the guidelines below to perform the task:
1. Examine the web page to understand the the domain of the web page.
2. Brainstorm 8 task categories that could be performed the website. Be creative.
3. For each task category, propose a concrete task that has this web page as one of its steps. You want the concrete task to be unambiguous and clear so that no further clarification is needed to perform the task.
4. Given a concrete task, you are ask to come up with the past actions that leads to the current page, as well as the next action.
* Requirement for past actions: You should write down each past action in the details. You want to group all actions into multiple sub-sections, each section corresponds to a logical and actionable sub-task. The next action could start with a new sub-task. You can omit the 'elemement_id' if they are not in the current page. There should only be one action at each step. DO NOT give goto() or new_tab() as first step.
* Requirement for next action: Provide the reasoning behind your past actions and the progress in completing the task. Also, describe your understanding of the current page and the concrete reason to execute the next action. If the action takes an element as the argument, it is important that you understand the role and the attributes of that element so that the action can be appropriately applied. Make sure to always include the 'element_id' in your next action if there is any. Any 'element_id' must come from the given Accessibility Tree.

## Format of the response
You are asked to provide the action sequence for task #1 with roughly 7 past actions; task #2 with roughly 0 past actions; task #4 with roughly 6 past actions; task #5 with roughly 4 past actions; task #6 with roughly 10 past actions. Your answer should follow the following format:

<Analysis and understanding about the domain and the concrete content of the web page>
<The list of 8 creative task categories>
<The concrete tasks for task category #1 #2 #4 #5 #6. Remember, a concrete task needs to include concrete details so that no further clarification is required when performing the task. Use imperative sentences.>

```python
# task: <repeat concrete task #1>

# ———————————
# past actions (history)
# sub-task 1: <sub-task description>
# step 1: <step description>
<action>
# step 2: <step description>
<action>
# sub-task 2: <sub-task description>
# step 3: <step description>
<action>
# step 4: <step description>
<action>
# sub-task 3: <sub-task description>
# step 5: <step description>
<action>
# step 6: <step description>
<action>
# sub-task 4: <sub-task description>
# step 7: <step description>
<action>

# ———————————

# next action
# step <index>: <summarize the progress so far and analyze the current state of the web page. Provide the concrete reason to perform the next action>
<action>
# step summary: <brief step description>
```
```python
# task: <repeat concrete task #2>
```

```
# ——————
# past actions (history)

# ——————
# sub-task <index>: <sub-task description>
# next action
# step <index>: <summarize the progress so far and analyze the current state of the web page. Provide the
concrete reason to perform the next action>
<action>
# step summary: <brief step description>
```
......
## The Accessibility Tree
[4812] link 'Shopbop Designer Fashion Brands'
    [4815] LineBreak
    [4819] LineBreak
[1742] LayoutTableCell ''
[1743] LayoutTableCell 'Amazon Warehouse Great Deals on Quality Used Products'
    [4822] link 'Amazon Warehouse Great Deals on Quality Used Products'
       [4825] LineBreak
       [4829] LineBreak
```

## G   Example Generated Trajectories from Random Observation

**Generated trajectories from random observation**

**Model Brainstorming**

The domain appears to be that of a major online retail website, likely Amazon, considering the references to "Prime," "Best Sellers," and "Amazon Basics," as well as services like "Magazine Subscriptions" and "Prime Video." The structure of the page includes a variety of product categories, a search textbox, and links to other services such as account sign-in and shopping cart. This suggests that users on this page can shop for a wide range of items, manage their account, or engage with support services.

**Creative Task Categories**: Below are eight creative task categories that could be performed on this type of website:

- Product Searching: Looking up products in a specific category or using a search query.
- Account Management: Signing into an account, creating a new account, or managing existing account details.
- Navigational Inquiry: Using the site's navigation to find different departments or special features.
- Customer Support: Accessing customer service features or disability support services.
- Deal Hunting: Searching for best sellers, deals of the day, or exclusive prime deals.
- Media Consumption: Navigating to and consuming digital media such as eBooks, Prime Video, or music.
- Educational Browsing: Exploring educational materials or subscribing to learning resources.
- Wishlist Management: Adding items to a wishlist or gift registry for future purchase.

Concrete Tasks The concrete tasks for task categories #1, #2, #6, #7, #8 are as follows:

**Concrete Task #1: Product Searching**
Find and search for popular fitness trackers within the "Sports & Outdoors" category.

```
# task: Find and search for popular fitness trackers within the
    "Sports & Outdoors" category.

# --------------------
# past actions (history)
# sub-task 1: Open the site and navigate to the product categories.
# step 1: The user opens the main page of the retail website.
go_forward()
# sub-task 2: Navigate to the "Sports & Outdoors" category.
# step 2: The user accesses the menu to view the category list.
click(element='Open Menu')
# step 3: The user browses through the categories.
scroll(down)
# step 4: The user selects the "Sports & Outdoors" category.
click(element='Sports & Outdoors', element_id=7964)

# --------------------
# next action
# step 5: The user now needs to search for fitness trackers within
    the selected category.
click_and_type(element='Search', content='fitness trackers',
    element_id=7657)
# step summary: Initiate a search for fitness trackers in the
    Sports & Outdoors category.
```

## Concrete Task #2: Account Management

Sign into your personal account to check on active subscriptions.

```
# task: Sign into your personal account to check on active
    subscriptions.

# --------------------
# past actions (history)

# --------------------
# sub-task 1: Sign into the account.
# next action
# step 1: The user is already on a page where they can initiate
    the sign-in process.
click_and_type(element='Search', content='myusername@example.com',
    element_id=7657)
# step summary: Try to sign in by using the search bar to enter
    the username.
```

## Concrete Task #6: Media Consumption

Navigate to Prime Video to browse the latest movie releases.
```

```
# task: Navigate to Prime Video to browse the latest movie
    releases.

# --------------------
# past actions (history)
# sub-task 1: Access the main page.
# step 1: The user starts by reaching the home page, where they
    can find various services.
go_back()
# step 2: The user considers different media options available on
    the site.
scroll(up)
# sub-task 2: Go to Prime Video.
# step 3: The user clicks on the "Prime Video" link.
click(element='Prime Video', element_id=7961)

# --------------------
# next action
# step 4: To search for the latest movie releases within Prime
    Video, the user needs to use the search function.
click_and_type(element='Search', content='latest movie releases',
    element_id=7657)
# step summary: Search for the latest movie releases on Prime
    Video.
```

**Concrete Task #7: Educational Browsing**

Subscribe to a magazine about technology within the "Magazine Subscriptions" category.

```
# task: Subscribe to a magazine about technology within the
    "Magazine Subscriptions" category.

# --------------------
# past actions (history)
# sub-task 1: Enter the site and locate the magazine section.
# step 1: The user starts on the home page where they can browse
    products.
go_forward()
# step 2: The user opens the categories menu to look for
    subscriptions.
click(element='Open Menu')
# step 3: The user scrolls through the menu to find the magazine
    subscriptions.
scroll(down)
# sub-task 2: Select the "Magazine Subscriptions" category.
# step 4: The user selects the "Magazine Subscriptions" option
    from the categories.
click(element='Magazine Subscriptions', element_id=7955)
# step 5: The user is now presented with different types of
    magazines but wants to find technology-related ones.
scroll(down)

# --------------------
# next action
# step 6: With the magazine subscriptions displayed, it is time to
    search for technology magazines specifically.
click_and_type(element='Search', content='technology magazine',
    element_id=7657)
# step summary: Search for technology magazines in the Magazine
    Subscriptions section.
```

**Concrete Task #8: Wishlist Management**

Add a popular sci-fi novel to your wishlist for future purchasing.

```
# task: Add a popular sci-fi novel to your wishlist for future
    purchasing.

# --------------------
# past actions (history)
# sub-task 1: Access the website's book section.
# step 1: The user has started on the homepage and is looking for
    books.
click(element='Books', element_id=6007)

# --------------------
# next action
# step 2: The user wants to find and add a sci-fi novel to their
    wishlist.
click_and_type(element='Search', content='popular sci-fi novels',
    element_id=7657)
# step summary: Begin the search for popular sci-fi novels to add
    to the wishlist.
```

# H   Prompt Example

> **User**
>
> OBSERVATION:
> [1744] link 'HP CB782A#ABA 640 Inkjet Fax Machine (Renewed)'
> [1749] StaticText '$279.49'
> [1757] button 'Add to Cart'
> [1760] button 'Add to Wish List'
> [1761] button 'Add to Compare'
> WEBSITE: onestopmarket
> OBJECTIVE: Add HP Inkjet Fax Machine to cart.
> PREVIOUS ACTION: None

> **Assistant**
>
> Let's think step-by-step.
> This page lists the information of HP Inkjet Fax Machine, which is the product identified in the objective.
> To add it to the cart, I will click the "Add to Cart" button.
> In summary, the next action I will perform is `click [1757]`.

> **User**
>
> OBSERVATION:
> Tab 1 (current): Curry Brand Shoes & Gear | Under Armour
> [4976] RootWebArea 'Curry Brand Shoes & Gear | Under Armour' focused: True
> [5059] link 'Skip to main content'
> [5060] StaticText 'Skip to main content'
> [5400] link 'FREE U.S. Shipping Orders $99+ & FREE Returns'
> [5066] link 'Need Help?'
> [5067] button 'US' hasPopup: menu expanded: False controls: country-dropdown
> [5422] button 'Welcome James' hasPopup: menu
> [5073] navigation 'Main Navigation'
> [5660] menuitem 'Men' hasPopup: menu expanded: False controls: men
> [5668] menuitem 'Curry' expanded: False
> [5672] textbox 'Search by Keyword or Item No.' hasPopup: listbox required: False controls: search-results
> [5099] button 'add to shopping bag'
> [5030] main ''
> WEBSITE: underarmour
> OBJECTIVE: Search the cheapest Curry brand unisex athletic shoes with the number 5.5, add to cart and checkout.
> PREVIOUS ACTION: # step 1: Click menuitem Curry on the main navigation bar, # step 2: click on link Shoes

# I   Case Study

We conducted a detailed examination of instances where `Synatra-CodeLlama` successfully completes tasks that `GPT-4-turbo` fails to accomplish on WebArena. Two key patterns emerged from these cases, which we outline here. Our analysis focuses on hard-level tasks requiring multiple steps to complete.

`Synatra-CodeLlama` **identifies details more effectively** In several scenarios, `Synatra-CodeLlama` outperforms `GPT-4-turbo` by detecting and utilizing detailed information such as hidden links and buttons on the page, whereas `GPT-4-turbo` focuses only on the most prominent components. In the scenario illustrated in Figure 6, both agents are tasked with displaying issues labeled as "bug." `GPT-4-turbo` immediately attempts to use the search box, but lacks the knowledge of how to search effectively in this case and guesses the keyword, which does not yield results. In contrast, `Synatra-CodeLlama` pays closer attention to the content displayed on the page and accurately identifies a link that lists all issues labeled as "bug."

| Tab 0 (current): Issues · Dashboard · GitLab

Dashboard · GitLab' focused: True
    [2374] link 'Skip to content'
    [2496] link 'Dashboard'
    [2548] button '' hasPopup: menu expanded: False
    [3669] textbox 'Search GitLab' required: False
    [3576] generic 'Use the shortcut key / to start a search'

(a lot of links and texts here)

    'Administrator'
    [2680] StaticText ' \xa0 '
    [2682] link 'type: bug 🐞'
    [3436] link 'Assigned to Byte Blaze' | **GPT-4-turbo**

Let's think step-by-step. This page has a textbox with ID [3669] to filter results. To get all the opened issues that report bugs, I can search for 'type: bug'. The textbox can filter the issues based on labels, and in GitLab, the label for bugs is 'type: bug'. The pressing of the 'Enter' key is necessary after typing to submit the search. In summary, the next action I will perform is ```type [3669] [type: bug] [1]``` | **Synatra-CodeLlama**

# step 2: Since we are already in the Issues section, we need to filter the issues by the 'Bug' label to list only the relevant issues.
click(element_id="2682")
# step summary: Filter issues by the 'Bug' label. |

Figure 6: An example task where `Synatra-CodeLlama` is successful by paying more attention to the displayed web page while `GPT-4-turbo` is not.

Similarly, in the scenario shown in Figure 7, both agents are tasked with "finding the number of commits Killian made on 3/5/2023." At this stage, both agents have already navigated to the page displaying Killian's commits, and they can now provide the answer by simply counting. However, `GPT-4-turbo` fails to register the displayed information: in its reasoning, it attempts to "view all the commits," even though they are already visible. In contrast, `Synatra-CodeLlama` pays closer attention to the information on the page and accurately identifies the relevant link showing Killian's commits.

| Tab 0 (current): Commits · main · The A11Y Project / a11yproject.com · GitLab
    [2892] textbox 'Search GitLab' required: False
    [5720] generic 'Use the shortcut key / to start a search'

(a lot of links and texts here)

    [4758] StaticText '1 commit'
    [5963] img "Kilian Valkhof's avatar"
    [5964] link 'Fix typos in the code examples (#1524)'
    [5968] link 'Kilian Valkhof' | **GPT-4-turbo**

Let's think step-by-step. I need to look for the commits made by Kilian on 3/5/2023. First, I need to find a way to view all the commits. The search box has an id of [2892], which I can use to find the commits. I can type "kilian commits on 3/5/2023" into this box and press enter. In summary, the next action I will perform is ```type [2892] [kilian commits on 3/5/2023] [1]``` | **Synatra-CodeLlama**

# step 3: Identify the commit made on 3/5/2023 by Kilian Valkhof and count the number of commits.
click(element_id="5964")
# step summary: Click on the commit message to see the details of the commit. |

Figure 7: An example task where `Synatra-CodeLlama` is successful paying more attention to the displayed web page while `GPT-4-turbo` is not.

`Synatra-CodeLlama` **demonstrates clearer logical flow between steps** In the example shown in Figure 8, both agents are filling out a form and have correctly entered the start and end dates. However, `GPT-4-turbo` incorrectly predicts that the next step is to re-enter the starting date, while `Synatra-CodeLlama` accurately interprets the status of the web page ("with the date range set to include the entirety of Q1 2023"), plans accordingly ("we are ready to generate the refund report"), and executes the correct action. This example illustrates that data generated by `Synatra` enhances

the model's capacity to accurately interpret and respond to context, leading to more efficient and reliable task execution in dynamic settings.

```
[1504] button 'Show Report'
.......
[2278] StaticText 'Date Used'
[2279] combobox 'Date Used'
hasPopup: menu expanded: False
[2283] combobox 'Period' hasPopup:
menu expanded: False
[2324] StaticText 'From'
[1281] textbox 'From *' required: True
    [2346] StaticText '01/01/2023'
[2286] button 'undefined \ue627'
[2333] StaticText 'To'
[1284] textbox 'To *' focused: True
required: True
    [2497] StaticText '03/31/2023'
[2289] button 'undefined \ue627'
[2292] StaticText 'Order Status'
[2293] combobox 'Order Status'
hasPopup: menu expanded: False
```

| GPT-4-turbo | Synatra-CodeLlama |
|---|---|
| Let's think step-by-step. The task is to generate a refund report for Q1, and the end date is 3/31/2023, which has been correctly entered in the "To" textbox. The "From" textbox with id [1281] should contain the date which marks the beginning of Q1, which is 01/01/2023. I will input this date into the "From" textbox. After filling in both the "From" and "To" dates, I will need to click the 'Show Report' button with id [1504] to generate the report. In summary, the next action I will perform is ` ```type [1281] [01/01/2023] [1]``` ` | # step 5: With the date range set to include the entirety of Q1 2023, we are ready to generate the refund report. The next action is to click the 'Show Report' button to display the data. click(element_id="1504") # step summary: Generate the refund report for Q1 2023. |

Figure 8: An example task where `Synatra-CodeLlama` is successful while `GPT-4-turbo` is not.

## J Data Selection from Random Observations in ClueWeb

We inspect a random sample of ClueWeb on its domain distribution. As shown in Figure 9, the majority of domains appear only once, making up 69.1% of all the web pages.

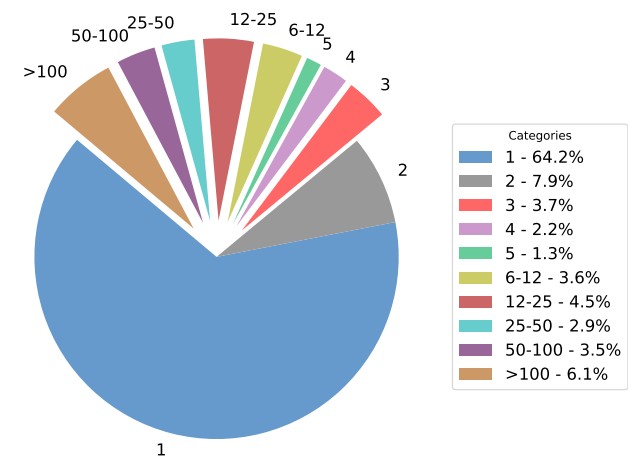

Figure 9: Frequencies of domains and proportion of each frequency in ClueWeb

## K Training Settings

The `CodeLlama` checkpoints are fine-tuned with A100 GPUs with deepspeed [7] acceleration framework. We set the context length to 4096 tokens. To train on $100k$ dataset, we train with 6 x A100

---

[7] https://github.com/microsoft/DeepSpeed

80G GPUs for about 40 hours. We use a batch size of 48, and learning rate of 4e-5. We use cosine annealing and a warm-up ratio of 0.03.

## L    Broader Impacts

The broader impacts of this work extend across several domains. Firstly, the approach has the potential to democratize the development of digital agents by making the training process more affordable and accessible. Organizations with limited resources can utilize existing public data to train competent agents without the need for expensive data collection efforts. This could lead to a more widespread adoption and innovation in AI applications, particularly in regions or sectors that previously could not afford such technology.

Secondly, by enabling digital agents to perform more complex tasks effectively, this work can significantly enhance productivity and efficiency in various industries. For example, customer service, online troubleshooting, and data management tasks could be accelerated, allowing human workers to focus on more creative or complex problem-solving tasks.

Furthermore, the technology has implications for accessibility, as it could help develop more intuitive and user-friendly interfaces for people with disabilities or those who are not tech-savvy. Agents trained with a diverse range of demonstrations can offer more personalized and context-aware assistance, improving user experience across digital platforms.

Lastly, the ethical and societal implications of this technology also constitute a critical area of impact. While the technology can lead to significant efficiencies and capabilities, it also raises questions about the potential for job displacement, and the need for robust guidelines to ensure that the deployment of such agents aligns with ethical standards. These broader impacts underscore the importance of interdisciplinary approaches to the development and governance of AI technologies, ensuring they contribute positively to society.

## M    Limitations

One major limitation to the work is the potential variability in the quality and relevance of the indirect knowledge sources. We attempted to mitigate this through use of high-quality sources such as WikiHow and broad sources such as ClueWeb with intelligent sampling strategies, but still the danger of unrepresentative data remains.

Another concern is the generalizability of the synthesized demonstrations. While the approach allows for the generation of a large volume of training data, the synthetic nature of these demonstrations may not fully capture the complexity and nuances of real human interactions with digital environments. As a result, agents trained on this data may still struggle with unexpected or less typical scenarios not covered in the training data.

Furthermore, there is the risk of overfitting to the specific formats and tasks represented in the indirect knowledge sources. If the diversity of these sources is limited, the agents may not develop the flexibility needed to handle a broad range of tasks across different platforms or environments.

Lastly, the reliance on large language models and complex synthesis processes might introduce significant computational costs and environmental impacts. The energy consumption and carbon footprint associated with training such large models are concerns that need to be addressed to ensure sustainable development in AI technologies. These limitations highlight the need for ongoing research, improved data curation methods, and the development of more robust models that can better generalize from synthetic training environments to real-world applications.

