# OpenReview forum: "Synatra: Turning Indirect Knowledge into Direct Demonstrations for Digital Agents at Scale"
_NeurIPS.cc/2024/Conference — NeurIPS 2024 poster_

### Official Review · Reviewer_UgDQ · 2024-07-03

**Soundness:** 3
**Presentation:** 3
**Contribution:** 3
**Rating:** 6
**Confidence:** 4

**Summary:**

The paper proposes a method for synthetically creating a dataset of agent trajectories for navigating websites. They propose using annotated programs, specifically python code interleaved with comments, as a representation of task and motion planning trajectories, where tasks are described in natural language in the comments, and actions are python API calls to the environment. A dataset of trajectories is created by applying GPT-4 to two sources of data. The first source is wikiHow, which contains ungrounded high-level tutorials. GPT-4 is used to generate plausible trajectories consistent with the tutorial. The second source is ClueWeb, which contains HTML snapshots of real websites. GPT-4 is again used to generate plausible trajectories consistent with the website. The paper finds that the resultant dataset of trajectories, when used to finetune CodeLlama-instruct-7b, outperforms open-weight models of similar sizes on Mind2Web, MiniWoB++, and WebArena. This result demonstrates that finetuning on the dataset transfers well to tasks.

**Strengths:**

- The paper is well-motivated, and for some sections clearly written
- Creative usage of two different data sources, tutorials and snapshots, to complement each other
- Makes progress towards building competent LLM-agents
- Demonstrates that synthetic trajectories improve benchmark performances.

**Weaknesses:**

- Requires the use of GPT-4 to generate the synthetic data. While the paper demonstrates that using GPT-4 to generate training data can improve smaller open weight models, it's not clear if this can help push the SOTA.
- Unequal comparison with human annotation: Sec. 6.2 compares with the human annotated trajectories in Mind2Web. As acknowledged by the authors, Mind2Web tasks are very restricted and simple compared with the tasks in synatra. A fairer comparison would be to collect a smaller set of human annotations on roughly the same task distribution as synatra.
- 6.3 rather ad hoc

**Questions:**

- open-source vs open-weight. Be careful with the distinction! (e.g. L54)
- How would GPT-4 perform if trained on synatra? Is synatra limited to distilling agent-capability from stronger models to weaker models? What stopped you from finetuning GPT-4 or GPT-3.5 on synatra?
- In 6.3 you write that you "conducted a detailed examination" comparing synatra with GPT-4, but I can't seem to find the analysis.
- Given that the comparison with human annotation in 6.2 doesn't seem very fair, I'm not sure if you can claim in the abstract that "we show that the synthetic demonstrations can be more effective than a similar number of human demonstrations."
- To help answer the quality/quantity trade-off, one might look at how finetuning codellama does when varying the size of the dataset. As prior work [0] suggested, maybe only a small number of examples is needed, which pushes the trade-off towards obtaining smaller quantities of high-quality human-annotated data instead of large quantities of synthetic data.

[0] Zhou, C., et al. "LIMA: Less Is More for Alignment. arXiv, Article." arXiv preprint arXiv:2305.11206 (2023).

**Limitations:**

- Broader impacts adequately discusses potential unethical usage of LLM agents enabled by this work.
- Needs discussion of robustness of LLM agents and potential security risks posed by deploying them.
- Limitations of using synthetic data adequately discussed.

---

> ### Author Rebuttal · Authors · 2024-08-07
>
> Thank you for taking the time to review our paper! We are very happy to hear the encouraging words on our motivation, creative usage of data sources, and performance improvement. We believe the comments are resolvable during the rebuttal period. Please see our response below. We would be very happy to address additional questions during the discussion period if anything is unclear.
>
>
> > ### How could Synatra possibly help push SOTA?
> > ### Is synatra limited to distilling agent-capability from stronger models to weaker models?
> > ###  Varying the size of the dataset.
>
> This is a great question, we believe Synatra holds the potential of pushing SOTA when scaling up. We additionally train a CodeLLama-7b on 100$k$ synthetic data and the results on different size of synthetic data is below:
>
>
> | Training Data Size | Success Rate on WebArena|
> |--------------------|--------------------------------------------------|
> | GPT-3.5| 6.16 |
> | 9k | 4.56 |
> | 50k | 4.80 |
> | 100k | 6.28 |
>
> We can see that training on more synthetic data provides 1.48% improvement on WebArena, surpassing GPT-3.5. In addition, among the 51 tasks `Synatra-CodeLLama-7b` got correct, we found `GPT-4-turbo` failed on 57% of them. For instance, while `GPT-4` suffers from severe errors such as repeatedly typing the same content to the same input box (26.85%), this only happens 6.9% of the time for `Synatra-CodeLLama-7b`. This indicates our approach could assist models in performing tasks the data generation model (e.g., `GPT-4-turbo`) cannot accomplish.
>
> > ### Can we compare the model trained on smaller set of human annotations on roughly the same task distribution as synatra?
>
> We agree it will be interesting to have such experiment, however, as discussed in the introduction section, collecting human annotations of similar task distribution can be challenging. The main difficulty is on configuring the task prerequisites. For example, in Synatra, we have synthetic trajectories to cancel Amazon order. However, it requires an Amazon account with qualified order for cancellation for human trajectory collection. Such setup is *instance specific*. Hence, it is prohibitive to collect human annotations at scale as in Synatra.
> In the paper, we also note that “it is important to recognize that the quality of human demonstrations is presumably higher, but they require meticulous design and control during the data collection process.” Hopefully this could further clarify the challenges in human data collection, and highlight the simplicity of our proposed approach.
>
> > ### I'm not sure if you can claim in the abstract that "we show that the synthetic demonstrations can be more effective than a similar number of human demonstrations."
>
> We will soften the claim to only compare with human demonstration *collected in a dataset-agnostic way*, meaning that the data collection process is not meticulous design and control.
> > ### What stopped you from finetuning GPT-4 or GPT-3.5 on synatra?
>
> On the one hand, the cost of finetuning GPT-3.5 or GPT-4 on **89 millon** tokens for a few epochs is prohibitive for us. On the other hand, we hope to push the frontier of open-weight models. We have spent more efforts on scaling up the data size as discussed above, and we are working on scaling up the model to stronger open-weight models.
>
> > ### 6.3 rather ad hoc
>
> Thank you for pointing this out. We will add more quantitative study in this section and move the case study to Appendix in our camera ready version.
>
> > ### Open-souce -> Open-weight
>
> Thank you for the suggestion. We will update the paper to reflect the accurate wording.

---

> > ### Comment · Reviewer_UgDQ · 2024-08-09
> >
> > Thanks for the response. I am surprised by the large increase in WebArena success rate from 50k to 100k. Do you have error estimates for these figures?

---

> > > ### Author Response · Authors · 2024-08-12
> > >
> > > Thanks! The variance between runs is as deterministic as possible with the current compute, as we have set the temperature to zero in LLM inference and fixed random seed during training. To find out the variance due to different random subset of training data in the 50k sample case, we need to sample different sets of data and do multiple training runs, which is out of our compute budget.

---

> > > > ### Comment · Reviewer_UgDQ · 2024-08-12
> > > >
> > > > Tthanks for the clarification. I still believe error estimates are important particularly when the figures seem so close together. Nevertheless, I believe the work to be essentially sound, and have updated my score.

---

### Official Review · Reviewer_JodV · 2024-07-13

**Soundness:** 3
**Presentation:** 3
**Contribution:** 2
**Rating:** 5
**Confidence:** 5

**Summary:**

This paper proposes a demonstration generation approach for learning UI control by mixing two different sources. The first one uses GPT4 to generate scenarios, grounded actions, and observations using WikiHow plans which is initially filtered by GPT-3.5. Another data source is ClueWeb, form which the authors extract web pages, sample a segment for any page, and ask GPT4 to come up with scenarios and next actions. Both data sources are combined into a single dataset of 50K traces where a program format is used. A CodeLlama model is fine-tuned with the resulting dataset. Results show that it achieves comparably when compared to other baselines, including a basic GPT-based prompting method. Ablation analysis shows that program format is important, both data mixtures add value, and proposed dataset transfer to other benchmarks better than Mind2Web traces.

**Strengths:**

The paper proposes a data synthesis approach for training UI agents. It combines WikiHow plans and actual observations with the strength of GPT-based models to fill in the gaps. the approach is agnostic to any domain and can serve as a good initialization.

**Weaknesses:**

My main concerns are weak baselines and lack of some others.

1. There are many baselines missing from the comparison and available baselines are weak. For example, HTML-T5 [11] achieves 49.7 and MindAct [6] achieves 30.9 on Mind2Web dataset. Performance on Miniwob++ reaches up to 99.2% [A]. There are many other baselines including prompting and fine-tuning based ones with different pre-training that should be included.

2. Can you further fine-tune on demonstrations from available benchmarks? While a semi-supervised approach is helpful, how much does it improve when human demonstrations added?

3. You also mentioned that a unique "id" is assigned to target elements in your training demonstrations. Given that the actions condition on the current state, what prevents the model from overfitting to this unique id when detecting salient elements?

4. You mention that you use viewports in Mind2Web benchmark to fit into the context window. But to extract viewports, you need to know where the ground truth element resides in the DOM (or accessibility tree) or you assume you have access to an oracle scrolling agent. I think both cases are different from your baselines and suggests leakage of test data.

5. When you use NL format rather than programs, how do you ground generated actions to executable environment actions?

6. What is the performance of the model trained with Mind2Web demonstrations on Mind2Web test sets? When tested on other benchmarks, how do you ground the action on the particular benchmark?

7. It is not clear if the claim that collected demonstrations have a similar distribution as Mind2Web as they have fatter tails.

8. Text and legend in Figure-2 are not readable.

A. Longtao Zheng, Rundong Wang, and Bo An. Synapse: Leveraging few-shot exemplars for human level computer control. arXiv preprint arXiv:2306.07863, 2023.

**Questions:**

Please see my above concerns for particular questions.

**Limitations:**

Limitations of the work are addressed.

---

> ### Author Rebuttal · Authors · 2024-08-07
>
> Thank you for taking the time to review our paper! We are very happy to hear encouraging words on our domain agnostic data synthesis approach that fills in the current gap. We believe the comments are resolvable during the rebuttal period. Please see our response below. And we would be very happy to address additional questions during the discussion period if anything is unclear.
>
> > ## Comparison with baselines
>
> > ### Comparison with LLMs finetuned with direct demonstrations (human annotated)
>
> ***Please Refer to the same section in global rebuttal due to character limit.***
>
> > ### Comparison with approaches that achieves high performance on MiniWoB++ or Mind2Web
>
> Thank you for bringing this up, we are aware of the many models and methods, we did not include them as baselines because of the following reasons:
> * Many approaches incorporate dataset-specific designs, making them challenging to apply to generic web-based tasks like WebArena. For instance, MindAct trained only with Mind2Web training data, as we can see with `Mind2Web-CodeLlama-7b`, while the model achieves strong performance on Mind2Web test set, it has limited performance on all held-out datasets. Similarly, Synapse uses trajectories as examples in the context; its prompting schema heavily relies on in-domain examples for MiniWoB++ and Mind2Web. In these datasets, the trajectories are short and can fit within the context window, and they offer relatively large training data on similar tasks or websites. However, in WebArena, trajectories can easily exceed the context length, and there are no provided training trajectories.
> * Many approaches are orthogonal to our data synthesis approach. The key component in MindAct is the ranking model that reduces a full HTML to the top-k potentially relevant elements, to remove irrelevant context. It is possible to apply a similar two-stage pipeline to our approach, instead of feeding the full AXtree into the model, we can first perform a reranking to select a subset of nodes in the AXtree instead. Alternatively, we can simply generate key elements rather than the full HTML as the observation.
> * Some approaches do not have public access. At the time we wrote the response, HTML-T5 is not open-sourced yet.
> We will incorporate the additional experiments to our updated draft. We are happy to investigate other approaches if we missed any.
>
>
> > ## Can you further fine-tune on demonstrations from available benchmarks? How much does it improve when human demonstrations added?
>
> Finetuning with both human demonstrations and our synthetic data is straightforward because our data design is general for web-based tasks. Further finetuning the model on high-quality human demonstrations can presumably bring significant gain. However, such data is scarce, and we hope our work can encourage the creation of more high-quality data.  In addition, our experiment shows that adding human demonstrations that are collected without meticulous control may not improve the model’s performance on general web-based tasks. As shown in the table in global rebuttal, simply training the model on human demonstrations collected in Mind2Web results in very limited performance on the two held-out sets. We also experiment with adding Mind2Web data into our synthetic data, and we also see a similar overfitting trend.
>
> > ## Could assigning unique “id” result in overfitting when detecting salient elements?
>
> We peformed a post processing to replace the unique ID with a **random** integer number in the final data. Hence, the model will not overfit to any specific ID.
> > ## You mention that you use viewports in Mind2Web benchmark to fit into the context window. But to extract viewports, you need to know where the ground truth element resides in the DOM (or accessibility tree) or you assume you have access to an oracle scrolling agent. I think both cases are different from your baselines and suggests leakage of test data.
>
> The main reason of doing this processing is to make sure the web page observation can fit into the context window of a model. We use the same processed web page to all baselines and our approach. Hence it is a fair comparison.
>
> > ## When you use NL format rather than programs, how do you ground generated actions to executable environment actions?
>
> The NL format we uses still requires the generated action to be in set formats. Here is an example:
>
> Past actions:
> 1. click [4904] where 4904 is img calendar wheel with marker
>
> For actions to be parsed accurately, they still need to be in the format such as click [xxx]
>
> > ## Performance of the model trained with Mind2Web demonstrations on Mind2Web test sets?
>
> The performance of the model trained with Mind2Web demonstrations on Mind2Web test sets is as followed:
> | Set | Step Accuracy |
> |--------------|-------------------------------|
>  | M2W-cross task | 0.47 |
>  | M2W-cross website | 0.34 |
>  | M2W-cross domain | 0.37 |
>
> > ## When tested on other benchmarks, how do you ground the action on the particular benchmark?
>
> We follow WebArena to represent a web page as an accessibility tree with ID at the beginning of each tree node. This can be done by processing the HTML in the three benchmarks. In this way, we can refer to any element with its corresponding ID such as `click(element_id=123)`.
>
> > ## It is not clear if the claim that collected demonstrations have a similar distribution as Mind2Web as they have fatter tails.
>
> Sorry for the confusion. Our goal was to demonstrate that our approach allows us to freely fit our generated trajectories to arbitrary distributions, such as Mind2Web. We will update Section 4 to clarify this.
>
> > ## Text and legend in Figure-2 are not readable.
>
> We will make sure to update the figures in the camera ready version.

---

### Official Review · Reviewer_Gani · 2024-07-16

**Soundness:** 3
**Presentation:** 3
**Contribution:** 3
**Rating:** 6
**Confidence:** 4

**Summary:**

This paper investigates turning indirect knowledge from the internet (such tutorial) to the direct knowledge (in the form of textual demonstrations) for LLMs. The algorithm design and the experiments are based on the web browser domain, which is a general interface to various web applications. The proposed method, Synatra, handles two types of knowledge source: tutorial articles and HTML snapshots, and presents corresponding demonstration synthesis approach. Such knowledge is then transferred to the LLMs by fine-tuning the LLMs on the synthesis data. Experiments show that Synatra can improve performance of CodeLlama-instruct-7b, outperforming open-sourced/fine-tuned models.

**Strengths:**

Overall, I think the paper is well-motivated. There are many tutorials available on the internet. It is interesting to explore how to effectively utilizing such knowledge.

The method is simple and effective, while the writing is easy to follow.

**Weaknesses:**

I have some concerns about the experiments.

1. The comparison experiment is a bit unfair to me, for several reasons:
- There are some RAG based methods to be considered, as this can be regarded as a knowledge retrieval problem.
- Much of the baselines are not fine-tuned. Although fine-tuned with interactive data, the data in AgentLM, CodeActAgent is not out-of-the-box like Synatra. Maybe the authors can consider baselines that fine-tune LLMs with more direct demonstrations.
- In Synatra, the demonstration is processed and generated by GPT-4. These high quality data further improves the performance of  Synatra.

2. There lacks some important details about the method, e.g., how is the LLM fine-tuned and the training hyper-parameters. Besides, the configurations about baselines and evaluation dataset should be detailed.

**Questions:**

1. How is success rate calculated in these web tasks?
2. In Synatra, how is the LLM fine-tuned on the synthesis data?
3. What is the source of performance improvement? Based on the results in Figure 4 (right), Synatra clearly outperforms human annotation method. I do not think this is simply attributed to the coverage of the annotation data. Maybe it is because GPT-4 has been trained on these testing tasks?

**Limitations:**

Authors have discussed the limitations in the conclusion section.

---

> ### Author Rebuttal · Authors · 2024-08-07
>
> Thank you for taking the time to review our paper! We are very happy to hear encouraging words about our motivation and effectiveness of our method. We believe the comments are resolvable during the rebuttal period. Please see our response below. We would be very happy to address additional questions during the discussion period if anything is unclear.
>
> > ### Comparison with RAG-based approach
>
> Thanks for pointing out this. We agree that the RAG-based approach is an alternative method of utilizing existing knowledge. Below is our comparison with the RAG-based approach. Specifically, we used the same collection of tutorials as the retrieval pool and employed cosine similarity of semantic embeddings of task descriptions to find the three closest tutorials. These tutorials were then included as additional context when prompting LLama3.1-instruct-8B to complete the tasks. Note that many RAG methods rely on *benchmark specific in-context examples*, we *do not* use any in this case. The results on WebArena are shown below:
> | Model                       | WebArena |
> |----------------------------|-------------------------|
> | RAG-Llama3.1-Instruct-8b  | 4.20 |
> | Synatra-CodeLlama-7b                 |  **4.80**           |
> We found that adding these additional tutorials fail to match the performance of Synatra. Since these tutorials were retrieved from wikihow articles, it is likely that they have been part of the large pre-training data already. However, converting them into direct demonstrations supplies additional information on the actual task completion, and hence can further improve performance.
>
> > ### Comparison with LLMs finetuned with more direct demonstrations
>
> ***Please Refer to the same section in global rebuttal due to character limit.***
>
>
> > ### Details about the method, e.g., how is the LLM fine-tuned and the training hyper-parameters; the configurations about baselines and evaluation dataset
>
> We apologize for the missing details. The details of the training are listed in Appendix I. The CodeLlama checkpoints are fine-tuned with A100 GPUs with deepspeed acceleration framework. We set the context length to 4096 tokens. To train on 50k dataset, we train with 6 x A100 80G GPUs for about 20 hours. We use a batch size of 48, and learning rate of 5e-5. We use cosine annealing and a warm-up ratio of 0.03. We use supervised fine-tune with full-parameters. We decided on this rather than lora from early experiment comparing the two’s performance. We only calculate loss on the response of the agent, while the observation, task title, past histories etc. are masked in loss calculation.  We will update section 5 to properly refer to the corresponding section in appendix I. We will update the details of baseline and evaluation setup in section 5 as well. For WebArena, we evaluated with a temperature of 0 and default set of parameters. For MiniWoB++, we evaluated on a subset that is the same as what's used in AgentLM/CodeActAgen. We take the five run-average success rate. We re-ran all the baseline models with the same setting for fair comparison. We will update these details in section 5 and appendix. And we will also release our training/evaluation code.
>
> > ### How is success rate calculated in these web tasks?
>
> Both benchmarks provide validators to programmatically check if a task is completed or not inside their corresponding environments. For instance, if a task is to put a certain item into the shopping cart, a program check would be to verify if the said item is in the cart by the end of the trajectory.
>
>
> > ### What is the source of performance improvement?
>
> Our analysis shows that synthetic data teaches the model on multiple levels.
> First, it provides an understanding of basic task formats. Initially, the base model had issues such as repeating actions, interacting with nonexistent elements, or performing invalid actions not specified in the prompt. After training, these errors significantly decreased, approaching GPT-4 levels. The detailed breakdown is shown below:
> | Model | Click on Non-functional Elements (\%) | Repeated Typing Actions (\%) | Click on Non-existent Elements (\%) |
> |-----------------------------|---------------------------------------------------|--------------------------------------|-------------------------------------------------|
> | GPT-4-Turbo | 6.4 | 26.85 | 1.66 |
> | Synatra-CodeLlama-7b | 4.9 | 6.8 | 0.7 |
>
> Second, synthetic data enhances the model’s understanding of web pages. For instance, the base model correctly typed into a typable element (e.g, a input box) less than 75% of the time. Post fine-tuning, this accuracy improved to over 93%.
> Lastly, we qualitatively examined model’s planning on the task before and after training, we found that synthetic data yields more accurate task planning and hence improve task performance.
>
>
> > ### Does the improvement come from GPT-4 trained on the test tasks?
>
> It is highly unlikely that GPT-4 was trained on the exact test tasks. Even if GPT-4 encountered some of the test data during training, our approach does not rely on its prior knowledge of them. Our data synthesis process is entirely independent of the evaluation datasets, including WebArena, MiniWob, and Mind2web. We did not utilize any specific test tasks or testing environments, such as the websites in WebArena, during data generation. Instead, we leveraged GPT-4 to transform randomly sampled wikiHow articles, targeting general web-based tasks, and randomly sampled web pages. Given the vast volume of web content, it is improbable that the sampled pages would overlap with those seen during the test.

---

> > ### Comment · Reviewer_Gani · 2024-08-08
> > **Response to author response**
> >
> > Thanks for your detailed response. It seems that other reviewers also concern about the comparison and the reliance on GPT-4. I appreciate authors' additional results, addressing some of concerns regarding the comparison.
> >
> > In addition, I still think the source of performance improvement over human demonstration is unclear. Specifically, I am not convinced that demonstrations generated by GPT-4 are more effective than that of human. You mentioned 'basic task formats'. However, human can also provide demonstrations that conform to the task format. The advantage of proposed method is that it does not require human cost, but requiring a high-quality LLM for annotation.

---

> > > ### Author Response · Authors · 2024-08-10
> > >
> > > Thanks for the follow-up! First, we hope that by showing models trained on Synatra is better than GPT-4-turbo in terms of repeating actions, interacting with nonexistent elements, and performing invalid actions help address the concern that Synatra is only learning from GPT-4-turbo and not teaching models new information GPT-4-turbo doesn't know.
> > >
> > > Regarding the comparison with human annotation (Mind2Web in this case), we agree that this could be more clear. Here is our finding from our experiments: we trained our two models with the exact same setting, one with Mind2Web's training set and the other with Synatra. As shown here and in global rebuttal:
> > >
> > > | Model                        | Mind2Web| MiniWoB++ | WebArena |
> > > |-----------------------------|----------------------|-----------------|-------------------------|
> > > | Mind2Web-CodeLlama-7b | **39.33** (held-in) | 27.00 | 0.00 |
> > > | Synatra-CodeLlama-7b                 | 17.26    | **39.57**  | **4.80**           |
> > >
> > > we found that the model trained with Mind2Web's training set actually **outperforms** the others on Mind2Web's test set. But it is also true that it is not as good as a Synatra trained model in MiniWoB++ and WebArena. The reason for this difference is relatively intuitive: by using synthetic data pipeline, we are able to generate all of the actions in the action space, including stop(), go_back(), go_forward(), press(), goto(), etc, while the action space of the Mind2Web data set only has click(), and type(). The limited and biased action space played a key part in the low performance of Mind2Web-CodeLlama on WebArena. Additionally, websites' domain distribution could play a role too -- Mind2Web covers a somewhat limited number of sites, while Synatra has broader coverage. One other possible suggestion is potential format biases, but we have parsed the format of Mind2Web's training set to be exactly the same as Synatra, so we can rule this out as a factor. So the remaining plausible explanations are the differences in website domains and the types of actions represented in the dataset.
> > >
> > > We hope this helps clarify, and we will further improve the discussion in the final version of the paper!

---

> > > > ### Comment · Reviewer_Gani · 2024-08-13
> > > >
> > > > Thanks for your reply. I hold my opinion that the synthetic data quality can not outperform human demonstration. However, that is not the point of this paper. I believe how to leverage LLM knowledge is a promising direction. Thus I increase my score.

---

### Official Review · Reviewer_5MC7 · 2024-07-29

**Soundness:** 3
**Presentation:** 3
**Contribution:** 3
**Rating:** 6
**Confidence:** 2

**Summary:**

This paper studies the problem of dataset generation for the digital task in the context of LLM. The motivation is that the current data collection process often requires human annotation, which is very costly and not scalable. As a result, the paper proposes using large language models (LLMs) to generate a synthetic dataset based on indirect knowledge. The indirect knowledge often refers to tutorials designed for human consumption and observation trajectories that lack associated tasks and actions. In contrast, "direction" knowledge consists of complete states and actions. To generate synthetic data based on indirect knowledge, the paper proposes several methods: (1) writing a prompt to let LLMs imagine their missing actions or states given partial trajectories, and (2) randomly sampling the context from the external large-scale dataset such as webpages. Table 1 shows the results that the proposed approach achieves the best performance among the baselines.

**Strengths:**

1. The paper is easy to follow and read.
2. The method seems to solve the common lack of demonstration data issue in the current LLM fine-tuned paradigm.
3. The method seems to be novel.

**Weaknesses:**

1. The method falls into the category of the perpetual data machine, in which one uses LLMs to generate its own training dataset. In addition, the proposed method is similar to self-refine paper (https://arxiv.org/pdf/2303.17651), in which LLMs iteratively fine-tune itself based on the generated dataset.
2. Table 1 does not compare the proposed method to the existing methods that use their own dataset to fine-tune such as self-refine paper. For instance, the paper could add a baseline in which the model is fine-tuned based on the other iterative fine-tuned procedure on the same dataset used in the paper. This will further verify the proposed method

**Questions:**

1. I am curious to learn more about the author's thoughts on perpetual data machines, and the comparison to self-refine papers.
2. Please answer the weakness section

**Limitations:**

See the weakness section.

---

> ### Author Rebuttal · Authors · 2024-08-07
>
> Thank you for taking the time to review our paper! We are very happy to hear the encouraging words about the work being novel and addressing the lack of demonstrations in LLM finetuning for agentic tasks. We believe the comments are resolvable during the rebuttal period. Please see our response below. We would be very happy to address additional questions during the discussion period if anything is unclear.
>
> > ### The comparison with self-refine
>
>
> The core difference between our approach and self-refine is that we apply LLM refinement *for different purpose*. Our proposal is a *data synthesize approach* that leverage LLM to transfer non-demonstrations into direct demonstrations and further finetune a model on the synthesized data. Our goal is to enhance a model’s performance on computer tasks. On the other hand, self-refine focuses on refining a model's *response* to a given prompt, it is a prompting mechanism. Self-refine does not involve generating training data and finetuning.
>
> > ### Comparison with models finetuned on their own dataset
>
> *Also included in global rebuttal*
>
> Direct demonstrations for web-based tasks are scarce, resulting in a limited number of LLMs fine-tuned with such data. We made our best effort to survey or implement approaches that are fine-tuned with direct demonstrations. We include three additional baselines and the results are shown below:
> | Model                        | Mind2Web| MiniWoB++ | WebArena |
> |-----------------------------|----------------------|-----------------|-------------------------|
> | AutoWebGLM-7b-42k | - | - | 2.50 |
> | Mind2Web-CodeLlama-7b | **39.33** (held-in) | 27.00 | 0.00 |
> | RAG-Llama3.1-Instruct-8b | - | - | 4.20 |
> | Synatra-CodeLlama-7b                 | 17.26    | **39.57**  | **4.80**           |
>
> Here, `AutoWebGLM-7b-42k` is finetuned with their 42$k$ synthetic direct demonstrations. The result is taken from [1]. `Mind2Web-CodeLLama-7b` is finetuned with the Mind2Web training data of 9$k$ direct demonstrations. The finetuning is done by us with the same configurations as Synatra. For `RAG-Llama3.1-Instruct-8b`, we used the same collection of tutorials as the retrieval pool and employed cosine similarity of semantic embeddings of task descriptions to find the three closest tutorials. These tutorials were then included as additional context when prompting LLama3.1-instruct8B to complete the tasks. We aims to compare different ways of using existing knowledge with `RAG-Llama3.1-Instruct-8b`
> We can see Synatra achieves the best performance on all *held-out* datasets.
> We also note that finetuning datasets used in AgentLM, CodeActAgent include a reasonable amount of out-of-box web-based demonstrations. In fact, Mind2Web training set made up 66% of the examples in AgentLM’s training set. We will incorporate the additional experiments to our updated draft. We are happy to investigate other approaches if we missed any.
> [1] Lai et al. Autowebglm: Bootstrap and reinforce a large language model-based web navigating agent.
>
>
>
> > ### Discussion on perpetual data machine
>
> Creating perpetual data machine is an exciting and open research question. Within the scope of our work, we propose leveraging existing resources. Currently, massive text corpora are primarily used for next-token prediction during pretraining, yet each piece of text can embed more diverse information. As models continue to improve through iterative training, they may be able to extract additional information or transform knowledge to enhance their capabilities.

---

### Author Rebuttal · Authors · 2024-08-07

> ### Comparison with LLMs finetuned with direct demonstrations

Direct demonstrations for web-based tasks are scarce, resulting in a limited number of LLMs fine-tuned with such data. We made our best effort to survey or implement approaches that are fine-tuned with direct demonstrations. We include three additional baselines and the results are shown below:
| Model                        | Mind2Web| MiniWoB++ | WebArena |
|-----------------------------|----------------------|-----------------|-------------------------|
| AutoWebGLM-7b-42k | - | - | 2.50 |
| Mind2Web-CodeLlama-7b | **39.33** (held-in) | 27.00 | 0.00 |
| RAG-Llama3.1-Instruct-8b | - | - | 4.20 |
| Synatra-CodeLlama-7b                 | 17.26    | **39.57**  | **4.80**           |

Here, `AutoWebGLM-7b-42k` is finetuned with their 42$k$ synthetic direct demonstrations. The result is taken from [1]. `Mind2Web-CodeLLama-7b` is finetuned with the Mind2Web training data of 9$k$ direct demonstrations. The finetuning is done by us with the same configurations as Synatra. For `RAG-Llama3.1-Instruct-8b`, we used the same collection of tutorials as the retrieval pool and employed cosine similarity of semantic embeddings of task descriptions to find the three closest tutorials. These tutorials were then included as additional context when prompting LLama3.1-instruct-8B to complete the tasks. We aims to compare different ways of using existing knowledge with `RAG-Llama3.1-Instruct-8b`
We can see Synatra achieves the best performance on all *held-out* datasets.
We also note that finetuning datasets used in AgentLM, CodeActAgent include a reasonable amount of out-of-box web-based demonstrations. In fact, Mind2Web training set made up 66% of the examples in AgentLM’s training set.
[1] Lai et al. Autowebglm: Bootstrap and reinforce a large language model-based web navigating agent.

---

### Decision · Program_Chairs · 2024-09-25

**Decision:**

Accept (poster)

**Comment:**

The paper received acceptance ratings from all reviewers. The reviewers initially had some concerns such as the fairness of the comparisons, lack of some important details, missing baselines, and unequal comparisons with human annotations. The rebuttal addressed some of the concerns. However, the reviewers still have some concerns regarding the quality of synthetic data vs human data and lack of error estimates. Despite the concerns, they still maintained a positive rating. The AC checked the paper, the reviews, and the rebuttal and believes the paper proposes a valuable contribution to the community. Hence, the AC follows the recommendation of the reviewers and recommends acceptance.